# Differentiable Parsing And Visual Grounding of Verbal Instructions for Object Placement

**Zirui Zhao, Wee Sun Lee, and David Hsu**
National University of Singapore
{ziruiz, leews, dyhsu}@comp.nus.edu.sg

**Abstract:** Grounding spatial relations in natural language for object placing could have *ambiguity* and *compositionality* issues. To address the issues, we introduce PARAGON, a *PARsing And visual GrOuNding* framework for language-conditioned object placement. PARAGON leverages object-centric relational representations for the visual grounding of natural language. It parses language instructions into relations between objects and grounds those objects in visual scenes. A particle-based GNN then conducts relational reasoning between grounded objects for placement generation. PARAGON encodes all of those procedures into neural networks for end-to-end training. Our approach inherently integrates parsing-based methods into a probabilistic, data-driven framework. It is data-efficient and generalizable for learning compositional instructions, robust to noisy language inputs, and adapts to the uncertainty of ambiguous instructions.

## 1 Introduction

Human-robot-interaction tasks, such as object placement, navigation, and assembly, often require detailed descriptions with spatial relations. Natural language provides a rich and intuitive interface for human-robot-interaction tasks [1]. Therefore, managing to learn and ground language-described spatial relations enables robots to assist us better. This research focuses on object placement tasks instructed by natural language. Humans verbally instruct robots to pick up an object and put it to a specific place. The robot generates object placements conditioned by language description and visual observation. However, spatial relations in natural language can be *ambiguous* and *compositional*, causing issues in language grounding.

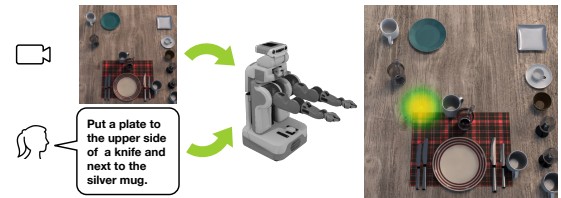

Figure 1. We aim to output placements based on visual and linguistic inputs. The presence of multiple semantically identical objects and omitted distance information cause difficulty for placement generation, and the compositional instructions increase the data required for learning.

We focus on two types of ambiguity: *positional ambiguity* and *referential ambiguity*. Positional ambiguity arises from the phenomenon that people tend to describe the directional relations without the precise distance (e.g., "to the left side"). In addition, we usually need a reference object to describe spatial relations (e.g., "next to the plate"). When placing an object next to reference objects, the connection between the reference object and placement is indirect, causing it to be difficult to learn object reference grounding and placement generation simultaneously. Furthermore, reference expressions of objects can be ambiguous, resulting in a reference expression being grounded to multiple semantically identical objects. It makes the distribution of correct placement multimodal. We refer to this issue as referential ambiguity.

The compositional structure of language-described spatial relations comes from the compositional nature of the visual scene and natural language. A complex scene contains multiple basic objects. To describe the desired state of a complex scene, one can compose many simple sentences for referents and their relations to form a complex language sentence (e.g., the instruction in Fig 1). This property increases the data required for learning compositional language instructions.

To address the issues, we introduce PARAGON, a *PARsing And visual GrOuNding* framework for language-conditioned object placement. The core idea behind PARAGON is to leverage structures

6th Conference on Robot Learning (CoRL 2022), Auckland, New Zealand.

in linguistic and visual inputs to extract object-centric relations for reasoning and placement generation, and encode those procedures in neural networks for end-to-end training. It learns to parse language into relational triplets from the grammatical structure. The triplets consist of subject, relation, and object phrases, e.g., ("plate", "upper side", "knife") in Fig 1. It then grounds the mentioned objects to the regions in the visual scene. A graph formed by grounded triplets is fed into a GNN for relational reasoning and generating placements. The GNN encodes a mean-field message passing algorithm to minimize the reverse KL-Divergence for the target distribution. We further develop a particle version of GNN to capture multimodal distributions.

PARAGON integrates parsing-based methods into a probabilistic, data-driven framework. It exhibits robustness from its data-driven property, as well as generalizability and data-efficiency from parsing. It adapts to the uncertainty of ambiguous instructions using particle-based probabilistic techniques. The experiments show that PARAGON outperforms the state-of-the-art method in language-conditioned object placement tasks in the presence of ambiguity and compositionality.

## 2  Related Work

Many works [2, 3, 4, 5, 6, 7, 8, 9, 10] developed solutions for language-instructed object manipulations of robotic systems. Our research focuses on object placement instructed by language. In contrast to picking [2, 3], which needs only a discriminative model to ground objects from reference expressions, placing [4, 5, 6, 7, 8] requires a generative model conditioned on the relational constraints of object placement. Specifically, it requires capturing complex relations between objects in natural language, grounding reference expression of objects, and generating placement that satisfies the relational constraints in the instructions.

Parsing-based robot instruction following [4, 5, 7, 11, 12, 13, 14, 15] parse natural language into formal representations using hand-crafted rules and grammatical structures. Those hand-crafted rules are generalizable but not robust to noisy language [1]. Among these studies, those focusing on placing [4, 5, 7] lack a decomposition mechanism for compositional instructions and assume perfect object grounding without considering referential ambiguity. Recently, [9, 10, 8] used sentence embeddings to learn a language-condition policy for robot instruction following, which are not data-efficient and hard to generalize to unseen compositional instructions. We follow [16] to integrate parsing-based methods into the data-driven framework. It is robust, data-efficient, and generalizable for learning compositional instructions.

PARAGON has a GNN for relational reasoning and placement generation, which encodes a mean-field inference algorithm similar to [17]. Moreover, our GNN uses particles for message passing to capture complex and multimodal distribution. The idea is to approximate a distribution as a set of particles [18], which provides strong expressiveness for complex and multimodal distribution. It is useful in robot perception [19, 20, 21], recurrent neural networks [22], and graphical models [23, 24]. Our approach employs this idea in GNN for particle-based message passing.

## 3  Overview

We focus on the language grounding for object placement in tabletop object manipulation tasks. In this task, scenes are composed of a finite set of 3D objects on a 2D tabletop. Humans give natural language $\ell \in \mathcal{L}$ to guide the robot to pick an object and put it at the desired position $\mathbf{x}_{tgt}^*$. The language instruction is denoted as a sequence $\ell = \{\omega_l\}_{1 \leq l \leq L}$ where $\omega_l$ is a word, e.g., in Fig.6, $\ell = \{\omega_1 = \text{put}, \omega_2 = \text{a}, \ldots\}$. A language instruction should contain a target object expression (e.g., "a plate") to specify the object to pick and express at least one spatial relation (e.g., "next to a silver mug") for placement description. The robot needs to find the distribution of the target object's placement $p(\mathbf{x}_{tgt}^*|\ell, \mathbf{z})$ conditioned on the language instruction $\ell$ and visual observation $\mathbf{z}$.

We propose PARAGON to solve the problem. It extracts object-centric relations from linguistic and visual inputs for relational reasoning and placement generation; it encodes those procedures in neural networks for end-to-end training. The pipeline of PARAGON is in Fig 2. PARAGON first uses the soft parsing module to convert language inputs "softly" into a set of relations, represented as triplets. A grounding module then aligns the mentioned objects in triplets with objects in the visual scenes. The triplets can form a graph by taking the objects as the nodes and relations as the edges.

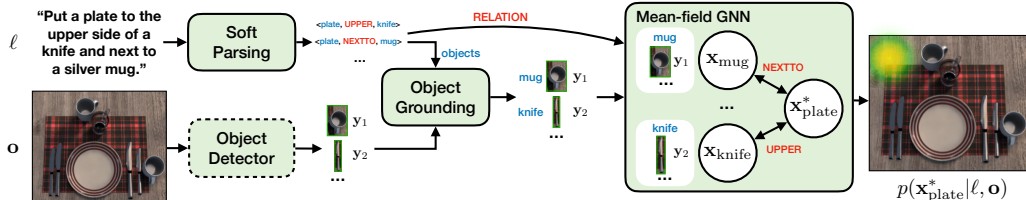

*Figure 2.* The overview of PARAGON. PARAGON uses soft parsing to represent language input as relations between objects. The grounding module then aligns the stated objects to the object-centric regions in the visual scene. An associated GNN conducts relational reasoning between grounded objects and outputs placement. This framework is trained end-to-end without labels for parsing and visual grounding.

The resulting graph is fed into a GNN for relational reasoning and generating placements. The GNN encodes a mean-field inference algorithm for a conditional random field depicting spatial relations in triplets. PARAGON is trained end-to-end to achieve the best overall performance for object placing without annotating parsing and object-grounding labels. See Appendix B for implementation details.

## 4 Soft Parsing

The soft parsing module is to extract spatial relations in complex language instructions for accurate placement generation. The pipeline is in Fig 4. Dependency trees capture the relations between words in natural language, which implicitly indicate the relations between the semantics those words express [25, 26]. Thus, we use a data-driven approach to explore the underlying semantic relations in the dependency tree for extracting relations represented as relational triplets. It takes linguistic input and outputs relational triplets, where the triplets' components are represented as embeddings. We first introduce the core concepts of triplets and dependency tree, then demonstrate the algorithm.

### 4.1 Preliminaries

**Triplets.** A triplet consists of two entities and their relation, representing a binary relation. Triplet provides a formal representation of knowledge expressed in natural language, which is widely applied in scene graph parsing [25], relation extraction [27], and knowledge graph [28]. The underlying assumption of representing natural language as triplets is that natural language rarely has higher-order relations, as humans mostly use binary relations in natural language [29]. For spatial relations, two triplets can represent ternary relations (e.g., "between A and B" equals "the right of A and left of B" sometimes). As such, it is sufficient to represent instructions as triplets for common object-placing purposes.

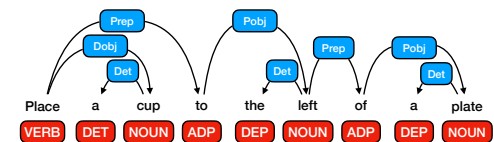

*Figure 3.* Dependency parsing takes language sequence as the input and outputs a tree structure. The blue blocks are dependency tags, while the red ones are part-of-speech tags. A part-of-speech tag categorizes words' correspondence with a particular part of speech, depending on the word's definition and context. Dependency tags mark two words relations in grammar, represented as Universal Dependency Relations.

**Dependency Tree.** A dependency tree (shown in Fig. 3) is a universal structure that examines the relationships between words in a phrase to determine its grammatical structure [30]. It uses part-of-speech tags to mark each word and dependency tags to mark the relations between two words. A part-of-speech tag [31] categorizes words' correspondence with a particular part of speech, depending on the word's definition and context, such as in Fig. 3, "cup" is a "NOUN". Dependency tags mark two words relations in grammar, represented as Universal Dependency Relations [32]. For example, in Fig. 3, the NOUN "cup" is the "direct object (Dobj)" of the VERB "place". Those relations are universal. A proper dependency tree relies on grammatically correct instructions, whereas noisy language sentences may result in imperfect dependency trees. Thus, we use the data-driven method to adapt to imperfect dependency trees.

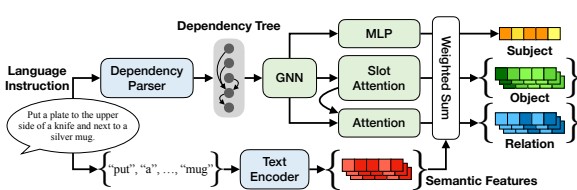

*Figure 4.* The pipeline of soft parsing module.

## 4.2 Method

To make the parsing differentiable, the soft parsing module "softens" the triplet as the attention to the words of linguistic inputs $\{a_{\gamma,l}\}_{1\leq l\leq L}, \sum_l a_{\gamma,l} = 1, \gamma \in \{\text{subj}, \text{obj}, \text{rel}\}$. We compute the embeddings of components in triplets by the attention-weighted sum of the word embeddings $\varphi_\gamma = \sum_l a_{\gamma,l} f_{\text{CLIP}}(\omega_l)$. The word embeddings are evaluated using pre-trained CLIP [33] $f_{\text{CLIP}}$. We use a GNN [34] to operate over the dependency tree from spaCy [35] to encode the structural information into the node features. For the GNN, the part-of-speech tags are node features, and the dependency tags are the edge attributes. Then, we feed the node features into three modules to evaluate each word's weights to indicate how much this word contributes to the components in a triplet. We use a single layer MLP to compute the attention of subject words $\{a_{\text{subj},l}\}_{1\leq l\leq L}$. Slot attention is a learnable module that can extract a set of task-dependent features called slots, which is used in discovering multiple objects in a visual scene. To detect multiple spatial relations in compositional instructions, we use slot attention module [36] to get $N$ features and attentions $\{a_{\text{obj},l}^n\}_{1\leq n\leq N, 1\leq l\leq L}$ of object phrases, where $N$ is the total number of triplets. We feed the features of object phrases as the query, and structural features as the key and value of attention, into a multi-head attention layer [37] to get the attention for corresponding relational phrases $\{a_{\text{rel},l}^n\}_{1\leq n\leq N, 1\leq l\leq L}$. We compute the attention-weighted sum of word embeddings from pre-trained language models to get the embeddings of $N$ possible relational triplets, denoted as $\Omega = \{\langle \varphi_{\text{subj}}, \varphi_{\text{rel}}^n, \varphi_{\text{obj}}^n \rangle\}_{n=1}^N$.

Ideally, each triplet should contain information about single objects and their relations. However, we discovered that the features of multiple objects and their relations could be entangled in one triplet. As the disentangled triplets can provide more precise essential information and be more accurate in recovering the sentence, we add sentence recovery to guide extracting disentangled triplets. We first compute the embedding of each token using attention: $\tilde{\varphi}_l = \sum_{n=1}^N \sum_{i\in\{\text{rel},\text{obj}\}} a_{i,l}^n \varphi_i^n + a_{\text{subj},l}\varphi_{\text{subj}}$. We then feed the embedding of tokens to an LSTM to produce the recovered word embeddings: $\tilde{\ell} = \{\tilde{\omega}_l\}_{l=1}^L = f_{\text{LSTM}}(\{\tilde{\varphi}_l\}_{l=1}^L)$. We minimize the L2 loss between generated sequence $l$ and $\tilde{l}$ as an auxiliary task to improve the soft parsing component.

# 5  Visual Grounding

After relation extraction, we ground the mentioned objects to the visual scenes and reason about their spatial relations for placement generation. We align the visual features of objects and the embeddings of their reference expressions into the same embedding space for object grounding. We then represent relations in conditional random fields and encode them into GNNs for reasoning and placement generation.

## 5.1  Object Reference Grounding

We align the object and subject phrase embeddings into the objects in the scene for visual grounding. Given the visual inputs, we use a pre-trained object detector (Mask-RCNN [38] or SPACE [39]) to get object bounding boxes $\{\mathbf{y}_m\}_{m=1}^M$ and encode their visual features $\{\mathbf{z}_m\}_{m=1}^M$ using a pre-trained CLIP [33]. We then project the object visual features $\mathbf{z}_m$ and linguistic features in triplets $\varphi_i$, $\varphi_i \in \{\varphi_{\text{subj}}\} \cup \{\varphi_{\text{obj}}^n\}_{n=1}^N$ into the same feature space via learnable projecting matrices $\Phi, \Psi$. We evaluate the cosine similarity $d_{\cos}(\cdot, \cdot)$ with a learnable scaling factor $\alpha$ between the visual and text feature to get the grounding belief: $b_m^i = \text{Softmax}_m(\alpha \cdot d_{\cos}(\Phi\mathbf{z}_m^\top, \Psi\varphi_i^\top))$. As such, the grounding results of object $i$ in the triplets is a set of object-centric regions with belief $\{(b_m^i, \mathbf{y}_m)\}_{m=1}^M$.

## 5.2  Relational Reasoning for Placement Generation

**Spatial relations in Conditional Random Field.** The triplets essentially build up a relational graph $\mathcal{G} = (\mathcal{V}, \mathcal{E})$ with objects as the nodes $\mathcal{V}$ and their spatial relations as the edges $\mathcal{E}$. We use the conditional random field (CRF) specified using $\mathcal{G}$ to represent the relations between positions of the context and target objects in language instruction. The variable for the vertices $\mathbf{X} = \{\mathbf{x}_i | \mathbf{x}_i \in \mathbb{R}^2\}_{i\in\mathcal{V}}$ denotes the grounded position of each context object and the placement position of the target object. We formulate the CRF as: $p(\mathbf{X}|\mathbf{z},\Omega) \propto \prod_{(i,j)\in\mathcal{E}} \psi_{ij}(\mathbf{x}_i,\mathbf{x}_j|r_{ij}) \prod_{i\in\mathcal{V}/\{\text{subj}\}} \phi_i(\mathbf{x}_i,\mathbf{z})$. $\psi_{ij}$ describes the spatial relations between two objects based on the spatial relation $r_{ij} \in \{\varphi_{\text{rel}}^n\}_{n=1}^N$. $\phi_i$ denotes the probability of the context object's position conditioned on observation, except for the

target object (subj phrase). It is because the grounded position of the target object is only useful for picking rather than placing. Mean-field variational inference approximates the CRF into a mean-field $p(\mathbf{X}|\mathbf{z}, \Omega) \approx q(\mathbf{X}) = \prod_{i \in \mathcal{V}} q_i(\mathbf{x}_i)$ and optimizes the reversed KL divergence $\mathrm{KL}(q||p)$ to converge to the multimodal distribution [40]. Mean-field $q$ factorizes all the variables to simplify the CRF for more efficient inference. The equation of Mean-field variational inference is: $\log q_i^t(\mathbf{x}_i) = c_i + \log \phi_i(\mathbf{x}_i, \mathbf{z}) + \sum_{j \in N(i)} \int_{\mathcal{X}} q_j^{t-1}(\mathbf{x}_j) \log \psi_{ij}(\mathbf{x}_i, \mathbf{x}_j|r_{ij}) \phi_j(\mathbf{x}_j, \mathbf{z}) d\mathbf{x}_j = c_i + m_{\mathrm{obs},i}^t(\mathbf{x}_i) + \sum_{j \in N(i)} m_{ji}^t(\mathbf{x}_i)$, where $q_i^t(\mathbf{x}_i)$ receives the message $m_{ji}^t(\mathbf{x}_i)$ from its neighboring nodes $j \in N(i)$ and the message from observed node $m_{\mathrm{obs},i}^t(\mathbf{x}_i)$, forming a message passing algorithm for $1 \le t \le T$. As such, the mean-field $q(\mathbf{X})$ will iteratively converge to $p(\mathbf{X}|\mathbf{o}, \Omega)$.

**Relational Reasoning in Mean-field GNN.** To represent complex spatial distributions conditioned on complex linguistic features, we follow [17] to map the spatial variables into high-dimensional feature spaces and learn an approximate message-passing function between these variables. It forms a GNN to conduct reasoning on the conditional random field by approximating the mean-field message passing algorithm. To handle multimodal output distributions better, we develop a method based on [22] to represent the factor of mean-field $q_i$ as particles $\{(h_{i,k}^t, w_{i,k}^t)\}$ in message passing, where each particle encodes the position of the corresponding object.

The GNN module takes the embeddings of spatial relations $r_{ij} = \varphi_{\mathrm{rel}}^n$ as the edge attributes to compute the message. We initialize the particles according to a normal distribution with uniform weights: $h_{i,k}^0 \sim \mathcal{N}(\mathbf{0}, \mathbf{I}), w_{i,k}^0 = 1/K$. In message passing, we uses a message network $f_\psi$ to output new message by the previous inputs $h_{j,k}^{t-1}$ and edge attributes $r_{ij}$: $m_{ji,k}^t = f_\psi(h_{j,k}^{t-1}, r_{ij})$. We then computes the embedding of observation by deepset [41]: $m_{\mathrm{obs},j}^t = f_\phi(\sum_{m=1}^M b_m^j f_{\mathrm{pos}}(\mathbf{y}_m))$, where $f_\phi$ and $f_{\mathrm{pos}}$ are MLPs. Following the form of mean-field inference, we compute the weights of the message $w_{ji,k}^t$ by the weights of particles and scores from observations computed by a weighting MLP $g_\phi$: $w_{ji,k}^t = \eta w_{j,k}^{t-1} \log g_\phi(h_{j,k}^{t-1}, m_{\mathrm{obs},j}^t)$.

Then, we aggregate the message from neighboring nodes and observations to compute the node features and beliefs. First, we sum up the messages and computes the new particles by an aggregate MLP $f_u$: $h_{i,k}^t = f_u(\sum_{j \in N(i)} m_{ji,k}^t + m_{\mathrm{obs},i}^t)$. We follow the mean-field inference to evaluate new particle weights by reweighting MLP $g_\phi$: $\log w_{i,k}^t = c + \log g_\phi(h_{i,k}^t, m_{\mathrm{obs},i}^t) + \sum_{j \in N(i)} w_{ji,k}^t$. Next, we use resampling to avoid particle degeneracy problems. Particle degeneracy refers to the weights of all but one particle being close to zero, resulting in extremely high variance. We resample after each iteration step to build a new set of particles with new weights: $\{(h_{i,k}'^t, w_{i,k}'^t)\}_{k=1}^K = \mathrm{SoftResamp}(\{(h_{i,k}^t, w_{i,k}^t)\}_{k=1}^K)$. However, this operation is not differentiable. Soft resampling [22] makes this process differentiable via sampling from a mixture of distribution $q(j) = \alpha w_j + (1 - \alpha)\frac{1}{K}$, and the new weights are evaluated using importance sampling, resulting in new belief: $w_k' = \frac{w_j}{(\alpha w_j + (1-\alpha)(1/K))}$. When $\alpha > 0$, soft resampling produces non-zero gradients for backpropagation. We take $\alpha = 0.5$ in training and $\alpha = 1.0$ in inference. When $t = T$, we decode the node features to get the final predictions of the target placement $\mathbf{x}_{i,k}^* = \pi_{\mathrm{dec}}(h_{i,k}'^T)$ and the weights $w_{i,k}^* = w_{i,k}'^T$.

# 6  End-to-end Training

We train the framework end-to-end by minimizing two objectives. One is the negative log-likelihood of dataset $\bar{\mathcal{X}}$: $\mathcal{J}_l = \sum_{\bar{\mathbf{x}} \in \bar{\mathcal{X}}} - \log \sum_k w_k^* \mathcal{N}(\bar{\mathbf{x}}; \mathbf{x}_k^*, \Sigma)$, where $\bar{\mathbf{x}} \in \bar{\mathcal{X}}$ is the labeled placement and the likelihood is a mixture of Gaussian represented by particles of target object $\{(\mathbf{x}_k^*, w_k^*)\}_{k=1}^K$. We also minimize the auxiliary loss for improving soft parsing: $\mathcal{J}_s = \sum_{\ell \in \bar{\mathcal{X}}} ||\ell - \tilde{\ell}||^2 = \sum_{\ell \in \bar{\mathcal{X}}} \sum_l (\omega_l - \tilde{\omega}_l)^2$. As such, the final objective is: $\mathcal{J} = \mathcal{J}_l + \lambda \mathcal{J}_s$, where $\lambda$ is a hyper-parameter.

# 7  Experiments

## 7.1  Experimental Setup

**Tabletop Dataset.** The tabletop dataset, as shown in Fig. 5, consists of 30K visually realistic scenes generated by PyBullet [42] and NVISII [43]. The objects are sampled from 48 objects with various shapes and colors. The images contain random lighting conditions, light reflections, and partial occlusions, reflecting the challenging language grounding in the real world. We use Mask-RCNN [38]

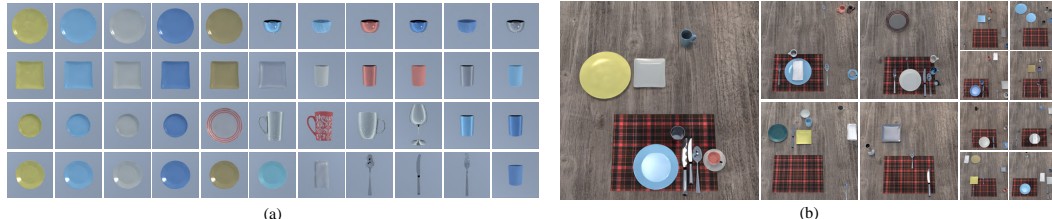

(a)  (b)

*Figure 5.* Images in the Tabletop dataset. The dataset contains more than 40 objects shown in (a) with randomly sampled materials and colors under random lighting conditions. The scenes shown in (b) are visually realistic and reflect the challenges in the real world.

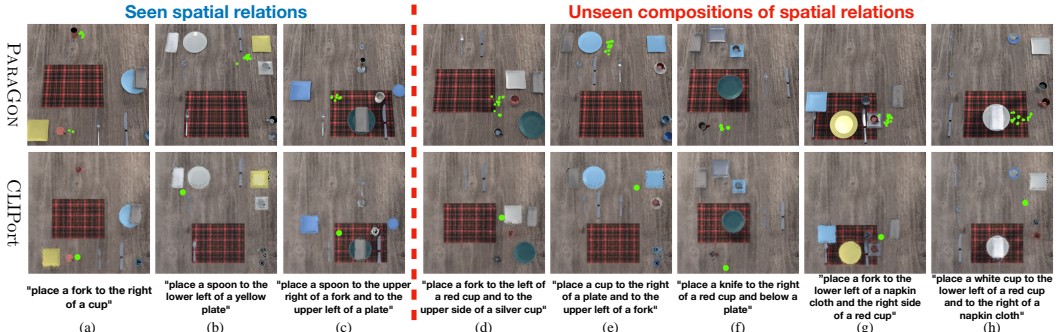

*Figure 6.* Demo of placement generation by PARAGON and CLIPort in an unseen visual scene. The green dots in the first row of images are the weighted particles of placement generated by PARAGON. The green dots in the second row are arg max of placement affordance generated by CLIPort. There are demos of instructions with one spatial relation (a), seen compositions of spatial relations (b, c), unseen compositions of spatial relations (d, e, f, g, h), and ambiguous case (a). The results show that PARAGON can reason complex spatial relations for suitable placements and exhibits a sense of generalizability to unseen compositions. CLIPort hardly generates correct placement for unseen compositional instructions, indicating its poor compositional generalizability.

for object detection. We also select three tasks in CLIPort's benchmark: placing objects inside a bowl or a box to test object reference grounding without positional ambiguity involved. Due to the lack of bounding box labels, we use the unsupervised object detector SPACE [39] for CLIPort's tasks. Other tasks of CLIPort's dataset focus on assembly or deformable object manipulation. They are not in the scope of our research.

We prepare *human-labeled instructions* to test the model with natural requests using human-provided natural language. Human-labeled instructions are from **bit.ly/TableSetData**. However, collecting enough human-labeled data is expensive, while synthesized data is easily generated. Hence, we use synthesized *structured language instructions* to train the models and fine-tune the model using human-labeled data. The training dataset of structured language instructions contains 20K instructions with single spatial relations and 20K with compositions of multiple spatial relations. Our testing dataset contains 15K instructions, including instructions with unseen compositions of seen spatial relations. We also have instructions containing ambiguous reference expressions, i.e., multiple objects are semantically identical to a reference expression. We prepare 9K human-labeled language instructions, where 7K instructions are for training, and the remaining 2K instructions are for testing. The human-labeled instructions are pre-collected from Mechanical Turk. We use image pairs to show the scene before and after an object is moved and let humans provide language requests for such object placement. The details of the dataset can be viewed at section A in the appendix.

**Evaluation Metric.** We evaluate the success rate of object placement, repeated 5 times. The successful placements should satisfy all the spatial constraints given in the language instructions. The placement should not be too far away from the reference objects with a threshold of 0.4 meters, which is approximated from the human-labeled dataset. The evaluated models are trained for 300K steps with a batch size of 1.

## 7.2 Results

We assess the performance of object placement by synthesized structured language instructions as well as human-labeled instructions, and compare our performance with CLIPort [8]. CLIPort has

a fully convolutional design with a two-stream architecture that handles spatial and semantic information. We first use three tasks from CLIPort's benchmark to test the models in grounding referred objects with referential ambiguity. We then use the Tabletop dataset to test language grounding with the presence of referential ambiguity, positional ambiguity, compositionality, as well as human-labeled noisy instructions.

**Object Reference Grounding.** In CLIPort's benchmark, spatial relations are simple as the placements are always inside a specific object. As such, learning placement generation equals learning object grounding. In these tasks, *packing-shapes* requires object grounding without referential ambiguity, *packing-google-objects* contains referential ambiguity for target objects, and *put-blocks-in-bowls* involves referential ambiguity for context objects. Table 1 reveals that both CLIPort and PARAGON have good performance in object grounding even with referential ambiguity.

*Table 1.* Results on CLIPort's Dataset: Success Rate (%)

| Tasks | packing-shapes | packing-google-objects | put-blocks-in-bowls |
|---|---|---|---|
| CLIPort [8] | 99.8 | 94.6 | 99.4 |
| PARAGON | 98.3 | 97.3 | 99.2 |

**Object Placing with Ambiguity.** The Tabletop dataset involves placing the object next to a context object, causing positional ambiguity. As the placement is not inside a specific object, object grounding no longer equals object placement. Table 2 suggests that PARAGON has a better performance with positional ambiguity. CLIPort uses convolutional architecture, which can capture strong local correlations. This inductive bias is useful if the placement is inside an object. However, positional ambiguity is only in weak agreement with the inductive bias as pixels far away can also be closely related, compromising the performance of CLIPort. PARAGON has an encoded graphical

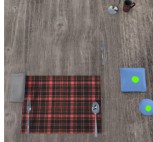 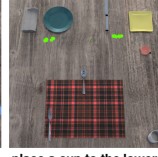 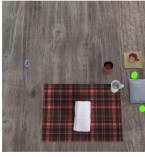 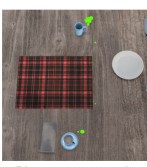

Place the red cup on a plate / place a cup to the lower side and left of a plate / Place a white cup below a square plate / Place a spoon to the upper right of a cup

*Figure 7.* Grounding instructions with referential ambiguity results in a multimodal distribution of object placement. We use particle-based GNN to capture the multimodality to adapt to referential ambiguity.

*Table 2.* Results on Tabletop Dataset: Success Rate (%)

| % of Training Data | Scene | No Ref ambiguity | | | With Ref Ambiguity | | |
|---|---|---|---|---|---|---|---|
| | Relation Type | Single | Comp | Comp* | Single | Comp | Comp* |
| 100% | CLIPort [8] | 73.2 | 69.1 | 59.5 | 71.4 | 68.2 | 51.7 |
| | PARAGON | **93.5** | **92.1** | **90.2** | **93.3** | **91.6** | **89.4** |
| 10% | CLIPort | 57.2 | 46.3 | 36.1 | 51.0 | 44.3 | 33.2 |
| | PARAGON | 89.7 | 88.2 | 88.0 | 89.9 | 87.6 | 87.3 |
| 2% | CLIPort | 39.7 | 29.1 | 29.5 | 33.9 | 28.5 | 22.9 |
| | PARAGON | 86.0 | 69.4 | 64.1 | 85.1 | 67.9 | 65.2 |

model for the spatial relations between objects to generate placement. It learns a distribution of the usual distance for placement from data and naturally adapts to positional ambiguity. We also test the performance of object placement with referential ambiguity (*With Ref Ambiguity* in Table 2), in which the scenes contain a few semantically identical objects. Referential ambiguity makes the correct placement non-unique, resulting in a multimodal distribution of correct placement. As shown in Table 2, PARAGON has a good performance with referential ambiguity. The core idea is to represent a distribution as a set of particles to capture multimodality and employ this idea in GNN for placement generation. Fig. 7 demonstrates that the GNN outputs a multimodal distribution when there is referential ambiguity.

**Compositionality.** Table 2 also reports the results on instructions with seen and unseen compositions of seen spatial relations (*Comp* and *Comp\** in Table 2, respectively), as well as the performance of the models trained with 2% and

*Table 3.* Human Instructions Results: Success Rate (%)

| | Fine-Tuned | | Not Fine-Tuned | |
|---|---|---|---|---|
| Method | PARAGON | CLIPort | PARAGON | CLIPort |
| Success Rate | 81.9 | 72.5 | 70.4 | 61.3 |

10% of the training data. The composition substantially increases the complexity of language comprehension and the data required for training. Table 2 shows that PARAGON is data-efficient in learning structured compositional instructions and can generalize to instructions with unseen compositions. It converts a complex language into a set of simple, structured relations represented as triplets to reduce complexity. Our approach operates on the grammatical structure of natural language that is generalizable to different semantic meanings. CLIPort uses single embeddings to

"memorize" seen compositions. It has poorer generalization in the presence of composition when training data is limited and does not generalize well to unseen compositions.

**Human-labeled Instructions.** We use human-labeled instructions to test natural human requests in noisy natural language. The instructions in this testing dataset may not be grammatically correct and contain unseen, noisy expressions for objects and spatial relations. The results are shown in Table 3, where *Fine-Tuned* means the models are fine-tuned by the human-labeled dataset, and *Not Fine-Tuned* models are trained only on structured instructions. The results show that PARAGON performs better on human-labeled instructions. PARAGON is data-driven and optimizes all the modules to adapt to imperfect, noisy linguistic data and extract useful relational information. PARAGON extracts relations from grammatical structures of instructions, which is highly generalizable and helps tackle unseen expressions. CLIPort is also data-driven but represents instructions as single embeddings that do not generalize well to unseen, noisy language expressions.

**Ablation Study.** We conduct an ablation study to assess each module's contribution in PARAGON. We design a variation of PARAGON without soft parsing, using single embeddings of language instructions to ground objects and represent relations. We also assess a version of PARAGON that uses a mean-field message passing neural network [17] without the use of particles, and test PARAGON without resampling. PARAGON uses pretrained CLIP to encode text and visual inputs for object reference grounding. To test the impact of using pre-trained models that are trained separately for vision and language, we replace CLIP in PARAGON with Visual Transformer [44] and BERT [45].

*Table 4.* Ablation Study: Success Rate (%)

| % of Training Data | Scene Relation Type | No Ref ambiguity | | | With Ref Ambiguity | | |
|---|---|---|---|---|---|---|---|
| | | Single | Comp | Comp* | Single | Comp | Comp* |
| 100% | PARAGON | **93.5** | 92.1 | 90.2 | **93.3** | **91.6** | 89.4 |
| | No Soft Parsing | 73.3 | 69.4 | 33.1 | 80.3 | 71.0 | 36.6 |
| | No Particle | 93.1 | **92.3** | **90.5** | 88.3 | 87.4 | 84.1 |
| | No Resampling | 93.2 | 92.0 | 81.2 | 86.3 | 90.7 | 78.9 |
| | PARAGON(ViT+Bert) | 91.1 | 92.0 | 88.3 | 90.8 | 91.4 | **89.9** |
| 10% | PARAGON | 89.7 | 88.2 | 88.0 | 89.9 | 87.6 | 87.3 |
| | PARAGON(ViT+Bert) | 88.4 | 86.0 | 85.2 | 87.7 | 86.5 | 85.4 |
| 2% | PARAGON | 86.0 | 69.4 | 64.1 | 85.1 | 67.9 | 65.2 |
| | PARAGON(ViT+Bert) | 70.1 | 65.3 | 58.6 | 69.8 | 61.1 | 52.7 |

We report ablation study results in Table 4. The results for *No Soft Parsing* demonstrate that soft parsing is essential in learning compositional instructions. It converts complex language sentences into simple phrases for objects and relations. The embeddings of those phrases have more straightforward semantic meanings than the entire sentence and are much easier for them to be grounded in the visual scene. As shown in the row of *No Particle*, using GNN without particles cannot capture multimodal distribution when referential ambiguity occurs and compromise the performance. The results in *No Resampling* indicate that resampling is helpful because particle degeneracy can occur without resampling, compromising the performance of capturing multimodal distributions with referential ambiguity. The last 4 rows in Table 4 show that PARAGON requires fewer data to obtain good results using pre-trained CLIP than pre-trained ViT+Bert. CLIP is pre-trained by a large dataset of image-caption pairs and is better for aligning language with visual features.

## 8 Conclusion and Limitation

PARAGON leverages object-centric representation for visual grounding of natural language. It integrates parsing into a probabilistic, data-driven framework for language-conditioned object placement. It tries to break the lines between parsing-based and embedding-based methods for language grounding, as well as combine the strength of rule-based and data-driven approaches. PARAGON reduces the complexity of embedding-based grounding by parsing complex sentences into simple structures, and learns generalizable parsing rules from data for robustness. Those combinations are beneficial and would facilitate future research. Our experiments reflect the difficulties of language grounding in real situations to show PARAGON's potential for real-world application.

PARAGON is limited by 2D object representations to ground 3D spatial relations situated by the 3D shape of objects, such as "lean". It also limits us to ground rotation instructions conditioned on the shape of objects, such as "rotate the fork to make it point to an apple". Leveraging 3D object representations would empower stronger language grounding skills and facilitate future research.

**Acknowledgments**

This research is supported in part by the National Research Foundation (NRF), Singapore and DSO National Laboratories under the AI Singapore Program (AISG Award No. AISG2-RP-2020-016) and the Agency of Science, Technology and Research, Singapore, under the National Robotics Program (Grant No. 192 25 00054).

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

# A  Tabletop Dataset

We extend the dataset from **bit.ly/TableSetData** with randomly sampled objects and generate placement from several sampled spatial relations. The dataset will be released in our website at **bit.ly/ParaGonProj**.

## A.1  Objects

We sample 50 dining objects with different classes, colors, and shapes, including forks, knives, spoons, plates, wine cups, water cups, and napkin cloths. The visualization of objects is in Fig. 5 (a).

## A.2  Language Expressions

Our dataset draws samples from commonly used spatial relations with various language expressions. The spatial relations includes left, right, front, behind, on top of, and their compositions. We also have ternary spatial relation "between". The language expressions for each spatial relations are illustrated in Table 5.

*Table 5.* Spatial relations and their language expressions in Tabletop Dataset.

| Spatial Relations | Language Expressions | Spatial Relations | Language Expressions |
|---|---|---|---|
| Left | "to the left of" 
 "to the left side of" 
 "to the left hand side of" | Front Right | "to the lower right of" 
 "to the lower right side of" 
 "to the lower right corner of" 
 "to the front right of" |
| Right | "to the right of" 
 "to the right side of" 
 "to the right hand side of" | Behind Left | "to the upper left of" 
 "to the upper left side of" 
 "to the upper left corner of" |
| Front | "in front of" 
 "to the lower side of" | Behind Right | "to the upper right of" 
 "to the upper right side of" 
 "to the upper right corner of" |
| Behind | "behind" 
 "to the upper side of" | on top of | "on" 
 "above" 
 "on top of" 
 "on the center of" |
| Front Left | "to the lower left of" 
 "to the lower left side of" 
 "to the lower left corner of" 
 "to the front left of" | | |

## A.3  Data Generation

### A.3.1  Instruction Generation

We generate the instructions according to language templates. The template for tasks with single spatial relation is illustrated below:

$$[\text{verb phrase}] + [\text{subject phrase}] \tag{1}$$
$$+ [\text{spatial relation phrase}] + [\text{object phrase}]. \tag{2}$$

If the language instruction contains compositional spatial relations, the template is:

$$[\text{verb phrase}] + [\text{subject phrase}] \tag{3}$$
$$+[\text{spatial relation phrase 1}] + [\text{object phrase 1}] \tag{4}$$
$$+\text{"and"} + [\text{spatial relation phrase 2}] + [\text{object phrase 2}]. \tag{5}$$

The spatial relations are sampled from Table 5, and the subject phrases and object phrases are sampled according to the object class, object colors, and shapes, such as "yellow square plate". We sample the verb phrases uniformly as random from "put", "place", and "move".

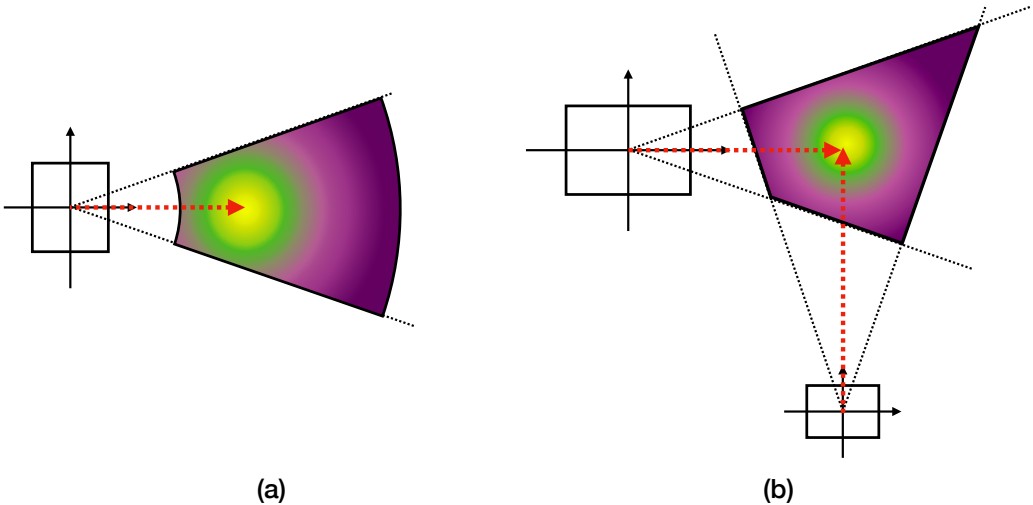

(a)                                (b)

*Figure 8.* Expert demonstration generation. The demonstration is sampled from a truncated Gaussian distribution. Case (a) illustrates a distribution of "left-hand side", where the highlighted area indicates a high probability. We truncate areas that derive from the left side and are too close to the reference object. Case (b) illustrates the composition of spatial relations "left-hand side" and "upper side", where the conjunction of both directions truncates the distribution.

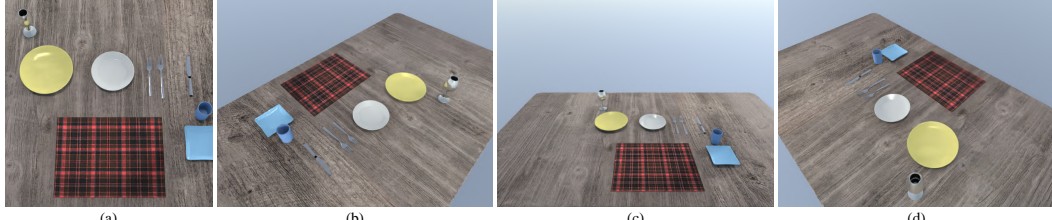

(a)           (b)           (c)           (d)

*Figure 9.* Raw observations captured by cameras in Tabletop Dataset.

### A.3.2 Image Generation

- **Camera Setup.** We use four cameras to capture the visual observation. Similar to the Raven simulator, there are three RGB-D cameras with resolutions of $640 \times 480$ that are placed on the rectangular table's top, left, and right, pointing towards the center (see (b), (c), and (d) in Figure 9). We also have an RGB camera with a resolution of $640 \times 640$ placed at the top of the table (see (a) in Figure 9 for the example picture).

- **Rendering.** We use NVISII as the rendering interface to get the photorealistic pictures of the visual scene. We randomly sample the position of the lights and their colors to add difficulties. In addition, due to the random lighting conditions, the visual scenes contain various light reflections, hampering the visual grounding of natural language.

### A.3.3 Expert Demonstration.

We provide a handcrafted policy for expert demonstration in placement generation. According to the ground truth position of objects in the simulator, we sample from a handcrafted Gaussian distribution with a specific direction as the placement generation policy. Specifically, the demonstration is sampled from a truncated Gaussian distribution. There are two examples shown in Fig. 8. In this figure, case (a) illustrates a distribution of "left-hand side", where the highlighted area indicates a high probability of placement. We truncate areas that derive from the left side and are too close to the reference object. Case (b) illustrates the composition of spatial relations "left-hand side" and "upper side", where the conjunction of both directions truncates the distribution. The mean of the Gaussian distribution is located at the intersection of the central axes of the spatial relations. We also use axis-aligned bounding boxes (AABB) of the target object and context objects to detect intersections and avoid object collision after generating the placement.

## A.4 Evaluation Critera

We evaluate the success rate of the object placing by examining whether the placement satisfies the spatial relational constraints in the natural language instructions. Specifically, we assess whether the generated particle is located at the truncated regions. For CLIPort, we take the `argmax` to get the generated placement. As ParaGon generates a set of particles as the samples of a placement distribution, we randomly sample from the particles ten times to evaluate the success rate.

# B  Implementation and Training Details

---
**Algorithm 1** Soft Parsing
---
**Input:** Instruction $\ell = \{\omega_l\}_{l=1}^{L}$, Text encoder $f_{\text{CLIP}}(\cdot) : \omega \to \mathbb{R}^{D_{\text{word}}}$, $K$

**Output:** Triplets: $\{(\varphi_{\text{subj}}, \varphi_{\text{obj}}^{(n)}, \varphi_{\text{rel}}^{(n)})\}_{n=1}^{N}$, Recovered instruction: $\tilde{\ell}$

1: $\tau \leftarrow \text{DepParser}(\ell)$
2: $\{\hbar_l\}_{l=1}^{L} \leftarrow \text{GNN}(\tau)$
3: $\{a_{\text{subj},l}\}_{l \leq L} \leftarrow \text{SoftMax}(\text{MLP}(\{\hbar_l\}_{l=1}^{L}))$
4: $\{a_{\text{obj},l}^{n}\}_{n \leq N, l \leq L} \leftarrow \text{SlotAttn}(\{\hbar_l\}_{l=1}^{L})$
5: $\hbar_{\text{obj}}^{n} \leftarrow \sum_l a_{\text{obj},l}^{n} \hbar_l$
6: $\{a_{\text{obj},l}^{n}\}_{n \leq N, l \leq L} \leftarrow \text{MHAttn}(\{\hbar_{\text{obj}}^{n}\}_{n=1}^{N}, \{\hbar_l\}_{l=1}^{L}, \{\hbar_l\}_{l=1}^{L})$
7: $\varphi_{\text{subj}} \leftarrow \sum_l a_{\text{subj},l} f_{\text{CLIP}}(\omega_l)$
8: $\varphi_{\gamma}^{n} \leftarrow \sum_l a_{\gamma,l}^{n} f_{\text{CLIP}}(\omega_l), \gamma \in \{\text{obj}, \text{rel}\}$
9: $\tilde{\varphi}_l = \sum_{n=1}^{N} \sum_{i \in \{\text{rel},\text{obj}\}} a_{i,l}^{n} \varphi_i^{n} + a_{\text{subj},l} \varphi_{\text{subj}}$.
10: $\tilde{\ell} = \text{LSTM}(\{\tilde{\varphi}_l\}_{l=1}^{L})$

---

## B.1  Soft Parsing

We illustrate the implementation details of the soft parsing module. The algorithm is shown in Algorithm 3.

### B.1.1  Structural Prior from Dependency Parsing

We use spaCy dependency parser with transformer backend to get the dependency tree of a given sentence. We use embeddings $v \in \mathbb{R}^{D_{\text{embed}}}$ to represent the part-of-speech (POS) tag and dependency (DEP) tag in the dependency tree. The message passing neural network uses MLP as the functions $f_m : \mathbb{R}^{D_{\text{embed}}} \times \mathbb{R}^{D_{\text{embed}}} \times \mathbb{R}^{D_{\text{embed}}} \to \mathbb{R}^{D_{\text{embed}}}$ and $f_u : \mathbb{R}^{D_{\text{embed}}} \times \mathbb{R}^{D_{\text{embed}}} \to \mathbb{R}^{D_{\text{embed}}}$. The POS tags are used as the initial node features and the DEP tags are the edge attributes. During the message passing, the node features and edge attributes are concatenated as the input of $f_m$ and $f_u$. Thus, the structural prior from the dependency tree is encoded in the output structural features of the message-passing neural network. In addition, the structure lacks the order of position of the tokens in the sentence. We use cosine positional encoding added in the output structural embeddings to encode the position of tokens in the sentence.

### B.1.2  Relation Extraction

The output structural features $\hbar_l$ from the message-passing neural network are used for relation extraction. We use attention to evaluate the normalized score of each token in the sentence from message passing for its contribution to the components in relational triplets. We use a single-layer MLP with softmax to compute the normalized score of each token contributing to the subject phrase. To discover potential object phrases, we use slot attention [36] from visual object discovery for unsupervised object phrase discovery. The slot attention algorithm is illustrated in Algorithm 2. After discovering the object phrases, we use a multi-head attention module to evaluate the relational phrases conditioned on the corresponding object phrases. The object structure embeddings from slot attention are used as the query, and the structural features from message passing as the key and value. After we compute the weights of each token in the sentence for the subject phrase, object phrase, and relational phrase, we use the expected embedding to represent the word embeddings for

the phrases in triplets. We use CLIP as the encoder to compute the word embeddings of tokens in the sentence.

### B.1.3 Hyper-parameters

The dimension of embedding is $D_{\text{embed}} = 64$, and the layer number of message passing neural network is 4. We use 3 iteration for slot attention, with slot number $N_{\text{slot}} = 4$, hidden dimension $D_{\text{slots}} = 128$ and $\epsilon = 1 \times 10^{-8}$. As for the multi-head attention module for relational phrase discovery, the number of heads is eight, and the embedding dimension is 64. Language encoder $\phi(\cdot)$ is trained CLIP (ViT-32/B), in which the word embedding dimension is $D_{\text{word}} = 512$.

---

**Algorithm 2** Slot Attention

---

**Input:** inputs $\in \mathbb{R}^{N \times D_{\text{inputs}}}$, slots $\sim \mathcal{N}(\mu, \text{diag}(\sigma)) \in \mathbb{R}^{K \times D_{\text{slots}}}$
**Layer params:** $k, q, v$ : linear projections for attention; GRU; MLP; LayerNorm($\times 3$)
1: inputs $\leftarrow$ LayerNorm(inputs)
2: **for** $t = 0, \ldots, T$ **do**
3:     slots_prev $\leftarrow$ slots
4:     slots $\leftarrow$ LayerNorm(slots)
5:     attn $\leftarrow$ SoftMax($\frac{1}{\sqrt{D}}k(\text{inputs})q(\text{slots})^\top$, axis = slots)
6:     Updates $\leftarrow$ WeightedMean(weights = attn + $\epsilon$, values = $v(\text{inputs})$)
7:     slots $\leftarrow$ GRU(state = slots_prev, inputs = updates)
8:     slots $\leftarrow$ slots + MLP(LayerNorm(slots))
9: **end for**
**Output:** slots

---

## B.2 Particle Mean-field GNN

The pseudo-code of particle mean-field GNN is illustrated in Algorithm 3.

### B.2.1 Functions in Particle Mean-field GNN

For the observation, we use masks of bounding boxes to represent the object positions and shapes in the observed image. We use a simple CNN model $f_{\text{pos}} : \mathbb{R}^{W_{\text{pos}} \times H_{\text{pos}}} \to \mathbb{R}^{D_{\text{pmpnn}}}$ to encode the positional feature of the bounding boxes detected in the scene. We then use an MLP to project the positional feature into embeddings. The particle proposal network $f_\psi : \mathbb{R}^{D_{\text{pmpnn}}} \times \mathbb{R}^{D_{\text{pmpnn}}} \to \mathbb{R}^{D_{\text{pmpnn}}}$ and aggregation network $f_u : \mathbb{R}^{D_{\text{pmpnn}}} \to \mathbb{R}^{D_{\text{pmpnn}}}$ are all single-layer MLP. The reweighting function of observation model $g_\phi : \mathbb{R}^{D_{\text{pmpnn}}} \times \mathbb{R}^{D_{\text{pmpnn}}} \to \mathbb{R}$ is also a MLP.

### B.2.2 Hyper-parameters

The layer number of particle-based mean-field GNN is $T = 8$. In message passing, we use embeddings with dimension $D_{\text{pmpnn}} = 128$. In soft resampling, we use $\alpha = 0.5$ for training and $\alpha = 1.0$ for inference. For visual grounding, the visual encoder is pre-trained CLIP (ViT-32/B).

## B.3 Training

We conduct supervised learning to train the model end-to-end. There are two components to the training objective. One is the negative log-likelihood of dataset $\bar{\mathcal{X}}$: $\mathcal{J}_l = \sum_{\bar{\mathbf{x}} \in \bar{\mathcal{X}}} -\log \sum_k w_k^* \mathcal{N}(\bar{\mathbf{x}}; \mathbf{x}_k^*, \Sigma)$, where $\bar{\mathbf{x}} \in \bar{\mathcal{X}}$ is the labeled placement and the likelihood is a mixture of Gaussian represented by each output particle of target object $\{(\mathbf{x}_k^*, w_k^*)\}_{k=1}^K$. We also minimize the auxiliary loss for improving soft parsing: $\mathcal{J}_s = \sum_{\ell \in \bar{\mathcal{X}}} ||\ell - \tilde{\ell}||^2 = \sum_{\ell \in \bar{\mathcal{X}}} \sum_l (\omega_l - \tilde{\omega}_l)^2$. As such, the final objective is: $\mathcal{J} = \mathcal{J}_l + \lambda \mathcal{J}_s$, where $\lambda$ is a hyper-parameter. The hyper-parameter $\lambda = 10.0$. For target object grounding, we use cross-entropy loss, where the labels for the loss function are the closest bounding box to the ground truth object position.

For training, we use Adam optimizer, and the learning rate is $1 \times 10^{-4}$. We train the model for 200K steps when the data is more significant than 1000 and 50K when trained by 300.

**Algorithm 3** Particle Mean-field GNN

---

**Input:** Number of layer $T$, number of particles $K$, detected object bounding boxes $\mathbf{B} = \{(\mathbf{z}_m, \mathbf{x}_m)\}_{m=1}^M$, relational graph $\mathcal{G} = (\mathcal{V}, \mathcal{E})$, edge attributes (spatial relation embeddings) $r_{ij}$

1: $h_{i,k}^{(0)} \sim \mathcal{N}(0,1)$, $w_{i,k}^{(0)} \leftarrow 1/K$, $t \leftarrow 0$          {Initialization}

2: **for** $t < T$ **do**

3:      **for** $(i,j) \in \mathcal{E}$ **do**          {Message Computing}

4:          $m_{ji,k}^t = f_\psi(h_{j,k}^{t-1}, r_{ij})$

5:          $m_{\text{obs},j}^t = f_\phi(\sum_{m=1}^M b_m^j f_{\text{pos}}(\mathbf{y}_m))$

6:          $w_{ji,k}^t = \eta w_{j,k}^{t-1} \log g_\phi(h_{j,k}^{t-1}, m_{\text{obs},j}^t)$

7:      **end for**

8:      **for** $i \in \mathcal{V}$ **do**          {Message Aggregation}

9:          $h_{i,k}^t = f_u(\sum_{j \in N(i)} m_{ji,k}^t + m_{\text{obs},i}^t)$

10:         $\log w_{i,k}^t = c + \log g_\phi(h_{i,k}^t, m_{\text{obs},i}^t) + \sum_{j \in N(i)} w_{ji,k}^t$

11:         $\{(h_{i,k}'^t, w_{i,k}'^t)\}_{k=1}^K \leftarrow \text{SoftResam}(\{(h_{i,k}^t, w_{i,k}^t)\}_{k=1}^K)$

12:      **end for**

13:      $t \leftarrow t + 1$

14: **end for**

15: $x_{i,k}^* = \pi_{\text{dec}}(h_{i,k}'^T), \forall k \leq K$          {Decode Placement}

**Output:** $\{(x_{i,k}^*, w_{i,k}'^T)\}_{k=1}^K$

---

## B.4  Baseline Implementation

### B.4.1  No Soft Parsing

This baseline model directly uses the embedding of the entire instruction sentence instead of the embeddings from triplets for object grounding and message passing in step 4 of Algorithm 3.

### B.4.2  No Particle

This baseline model does not have particles in the message passing, and the initially hidden features in step 1 of Algorithm 3 are vectors with zero values. It also does not have step 11 for soft parsing.

### B.4.3  No Resampling

The third baseline is built without soft resampling in step 11 of Algorithm 3.

### B.4.4  PARAGON(ViT + Bert)

This baseline model uses the ImageNet-pretrained ViT 32/B model for visual encoding and the pretrained Bert model for text encoding in visual grounding, representing the edge attribute $r_{ij}$.

