# OpenReview forum: "Differentiable Parsing and Visual Grounding of Verbal Instructions for Object Placement"
_robot-learning.org/CoRL/2022/Workshop/LangRob — LangRob 2022 Poster_

### Official Review · Reviewer_Lrkj · 2022-11-08
**Good paper**

**Rating:** 7
**Confidence:** 4

**Review:**

The paper presents novel method for grounding spatial relations in natural language for object placing, which also considers ambiguity. The method leverages soft parsing, pretrained CLIP features and a particle based graph neural network to achieve this.
The paper is mostly well written, though it could be more concise at some points and would benefit from a clear "these are our contributions" paragraph.
For example, the paper claims to be robust to noisy language inputs, however it is not clear how this is quantified, backed up or to which extent this. It looks like Table 3 should show this, but since the baseline also seems to do a reasonable job it is unclear if this should be a major claim of the paper. Also it would be good to show what some of those noisy language inputs are and how they diverge from the training set.
Besides, although I appreciate the ablation studies, I think that in order to better simulate real world conditions, there should be an ablation which shows the robustness with respect to visual noise, concretely, what happens if the detector does not detect all objects at test time. Since the method relies on having all objects detected to feed the GNN, it would be beneficial to test if still can do a reasonable if always a random node is missing as an ablation.
Finally, I would appreciate more details on how the training data was generated. Was it created with some oracle, was it teleoperated by humans? I ask this, because relational placements are very much subjective. If you ask a human to place x left of y, there is positional ambiguity as spatial relations do not uniquely identify a location on the table. I like that the paper describes this in the introduction, but it is unclear if the training data reflects this human multi-modality or it is just a heuristic where you sample positions from a gaussian or something. In the same line, what are your thoughts on asking help from the human for ambiguous instructions?

Typo at line 171: Object Refernece Grounding

---

### Official Review · Reviewer_BYCT · 2022-11-12
**Good paper, but a bit complicated**

**Rating:** 7
**Confidence:** 4

**Review:**

This paper presents ParaGon, a framework for grounding referring expressions with positional and referential ambiguities for tabletop rearrangement. ParaGon uses an object-detector to isolate objects, and a “soft-parser” to extract <obj1, relation, obj2> tuples. These detections and tuples are run through a Mean-Field GNN to output 2D pick-and-place actions.

Strengths
+ ParaGon achieves compelling results with respect to CLIPort, while being more data-efficient.
+ Experiments include a good set of baselines and ablation results.
+ The choice of dataset (Tabletop) seems appropriate.

Weaknesses and Suggestions:
- The overall approach is very complicated and hard to follow. Especially Page 4 and 5 go into so much detail that sometimes it’s hard to understand the motivation behind each design choice.
- While ParaGon outperforms CLIPort, it makes a lot of extra assumptions like pre-trained object-detectors and soft-parsing, which might hurt its general applicability to broader rearrangement tasks.
- Other works on the CLEVR dataset (and its variants) have extensively studied grounding complex spatial relationships. Perhaps benchmarking against those approaches would also be appropriate here.

---

### Decision · Program_Chairs · 2022-11-15

Accept (Poster)